# Treatment of Scaphoid Non-Unions with Custom-Made 3D-Printed Titanium Partial and Total Scaphoid Prostheses and Scaphoid Interosseous Ligament Reconstruction

**DOI:** 10.3390/healthcare11243123

**Published:** 2023-12-08

**Authors:** Alessio Cioffi, Giuseppe Rovere, Francesco Bosco, Ennio Sinno, Leonardo Stramazzo, Francesco Liuzza, Antonio Ziranu, Michele Romeo, Giulio Edoardo Vigni, Nicolò Galvano, Giulio Maccauro, Pasquale Farsetti, Mario Igor Rossello, Lawrence Camarda

**Affiliations:** 1Department of Orthopaedic Surgery (DICHIRONS), University of Palermo, 90133 Palermo, Italy; ale.cioffi@unipa.it (A.C.); stramazzoleonardo@gmail.com (L.S.); michele.romeo01@unipa.it (M.R.); giulioedoardovigni@unipa.it (G.E.V.); nicolo.galvano@unipa.it (N.G.); 2Department of Orthopaedics and Traumatology, Fondazione Policlinico Universitario A. Gemelli IRCCS, Università Cattolica del Sacro Cuore, 00168 Rome, Italy; giuseppe.rovere02@icatt.it (G.R.); francescoluzza@policlinicogemelli.it (F.L.); antonio.ziranu@policlinicogemelli.it (A.Z.); giulio.maccauro@policlinicogemelli.it (G.M.); 3Department of Clinical Science and Translational Medicine, Section of Orthopaedics and Traumatology, University of Rome “Tor Vergata”, 00133 Rome, Italy; faretti@uniroma2.it; 4Department of Orthopaedics and Traumatology, University of Turin, 10126 Turin, Italy; francesco.bosco@unito.it; 5Orthopaedic and Traumatology Department, S. Spirito Hospital, 00193 Rome, Italy; ennio.sinno@aslroma1.it; 6Hand Surgery Department “Renzo Mantero”, Ospedale San Paolo, 17100 Savona, Italy; info@chirurgiamanoossello.it

**Keywords:** custom-made, 3D, scaphoid, replacement, ligament reconstruction, SLIL, level of evidence, IV

## Abstract

Purpose: Treatment of scaphoid fracture sequelae is still an unsolved problem in hand surgery. Custom-made 3D-printed titanium partial and total scaphoid prosthesis and scaphoid interosseous ligament reconstruction (SLIL) are performed in cases of non-union and isolated aseptic necrosis of the proximal scaphoid pole and when it is impossible to save the scaphoid bone, respectively. This study aims to evaluate the clinical, functional and radiographic results after these two prosthesis implantations. Methods: Between January 2019 and July 2020, nine partial and ten total scaphoid prostheses were implanted using custom-made 3D-printed titanium implants. Evaluation criteria included carpal height ratio (CHR), radioscaphoid angle, wrist extension and flexion, radial deviation and ulnar deviation of the wrist, grip strength and pinch strength, Visual Analogue Scale (VAS), the Disabilities of Arm, Shoulder, and Hand (DASH) score, and the Patient-Rated Wrist Evaluation (PRWE). Results: Clinical, functional, and radiographic improvements were found in all outcomes analyzed for both patient groups. The VAS pain scale obtained the most remarkable improvement at the one-year follow-up. The results of the DASH scores and the PRWE were good, with a great rate of patient satisfaction at the end of the follow-up. SLIL reconstruction also provided excellent stability and prevented a mid-carpal bone collapse in the short- and medium-term follow-up. Conclusions: A custom-made 3D-printed titanium partial or total scaphoid prosthesis is a viable solution for patients with scaphoid non-union and necrosis or complete scaphoid destruction in whom previous conservative or surgical treatment has failed.

## 1. Introduction

The management of scaphoid fractures sequelae is one of the difficult issues in hand surgery. Delayed diagnosis of these injuries results in non-union, with or without necrosis. The progression of radioscaphoid osteoarthritis typically involves damage to the biomechanical structure of the radiocarpal joint. This condition may manifest as instability within the intercalated segments, characterized by dorsal intercalated segment instability (DISI) and volar intercalated segment instability (VISI), and can eventually lead to a total carpal collapse, specifically known as scaphoid non-union advanced collapse (SNAC) [1,2].

The scaphoid bone, located in the wrist, is the carpal bone that fractures most frequently, representing about 60% of all wrist fractures [1,2]. Typically, such a fracture results from a person falling onto an extended hand, which applies an excessive force of hyperextension to the wrist.

Scaphoid fractures may be initially difficult to diagnose on preliminary radiographs, which often seem unremarkable at first sight [3]. Computed topography (CT) scans and magnetic resonance imaging (MRI) are more sensitive and advanced imaging techniques that allow early diagnosis by identifying the rime fracture, bone oedema, and soft tissues involvement such as ligament or tendon lesion, etc. However, these imaging techniques may be costly and time consuming and are thus not always performed. For patients experiencing “snuffbox” tenderness in the scaphoid area after a wrist trauma, the recommended initial treatment typically involves immobilizing the wrist with a cast or brace. If tenderness persists, additional X-rays may be taken about two weeks post-injury [1,2]. There are several therapeutic options available for reconstructing the scaphoid or addressing and preventing subsequent complications, including bone grafts, partial or complete scaphoid removal combined with mid-carpal fusion, proximal row carpectomy (PRC), complete wrist replacement, or full wrist fusion [3,4,5,6]. Total prosthetic replacement of the scaphoid bone is an option when the bone cannot be preserved and if there is no evidence of mediocarpal or radiocarpal joint arthrosis or collapse [3,4,5,6]. On the other hand, partial prosthetic replacement is recommended in cases where a non-union has resulted in isolated aseptic necrosis of the scaphoid’s proximal pole [7]. The earliest scaphoid prosthesis was crafted from Vitallium in 1945, with subsequent developments including an acrylic version in 1950 and a silicone implant in 1962. The latter was responsible for many complications, such as “silicone synovitis”, which in some cases produced severe wrist degeneration [8,9]. Titanium prostheses were developed, bringing good performance with long-term control [10], but were never widespread use in clinical practice. Later, BioProfile^®^ (Tornier, Edina, Minnesota) introduced an adaptive proximal scaphoid implant (APSI) made of pyrocarbon, based on the concept of carpal adaptation to the implant, not anatomical reconstruction [11]. This prosthesis type has been popular due to the relatively straightforward technique, with good short- and medium-term results, but with frequent carpal collapse and implant penetration into neighboring bone segments in long-term follow-ups [12]. Since 2018, with powder technology and customized implantology development, it has been possible to construct “custom-made” total or partial scaphoid prostheses by 3D CT reconstructions of the individual wrist.

The criteria for proceeding with this surgical procedure are rigorous: it is reserved for cases where the scaphoid bone is too damaged for reconstruction using grafting methods, there is a demonstrated stability of the wrist and no SNAC wrist conditions as evidenced by measurements of carpal height and the angle between the radius and the scaphoid, and there are no degenerative changes in the radial facet of the scaphoid or other carpal bones [13,14]. The design of the new prostheses was specifically aimed at enabling the reconstruction of the scapholunate interosseous ligament (SLIL), which is crucial for maintaining the stability of the implant and the overall biomechanics of the wrist [7]. We think that 3D-printed titanium partial and total scaphoid prostheses and scaphoid interosseous ligament reconstruction could be a reliable surgical procedure for the treatment of scaphoid non-unions.

This study aims to evaluate the clinical, functional and radiographic results after implanting custom-made 3D-printed titanium partial and total scaphoid prostheses and reconstruction of SLIL.

## 2. Materials and Methods

### 2.1. Search Strategy

A retrospective study of a consecutive series of 9 partial and 10 total 3D-printed custom-made titanium scaphoid prostheses was performed. These prostheses were implanted by the same senior hand surgeon at the Hand Surgery Department of the San Paolo Hospital in Savona (Italy) between January 2019 and July 2020. Before surgery, all patients underwent magnetic resonance imaging (MRI), and computed tomography (CT) scans. The scans obtained were then sent to the ADLER^®^ manufacturer to create patient-specific 3D implants.

### 2.2. Inclusion and Exclusion Criteria

Patients treated with a partial scaphoid prosthesis had non-union at the proximal pole of the scaphoid with necrosis. This diagnosis was reached after CT scan and MRI, after 3 years in media from the injury. They had failed conservative or previous surgical treatment (Figure 1 and Figure 2). Patients undergoing total prosthesis had scaphoid non-union and irreparable bone damage due to diagnostic error or surgical treatment failure. Of the patients with a partial prostheses, 9 out of 9 had no treatment ab initio. Of the 10 patients with total prostheses, 5 had no treatment (misdiagnosed), 3 had a plaster cast for 2 months, 1 was treated with k-wires removed after 6 weeks, and 1 was treated with a ridge graft according to the Matti–Russe technique. Patients with wrist instability, SNAC, or scaphotrapeziotrapezoidal (STT) arthritis were excluded.

### 2.3. Institutional Database, Data Collection, and Patients Setting

We gathered personal and demographic information from everyone participating in the research. We reviewed hospital documents to ascertain the trauma specifics, any immediate additional injuries, and data from during and around the time of surgery. We recorded any local or systemic issues that arose from the surgical procedure. Standard diagnostic procedures for patients directed to the Hand Surgery Department at San Paolo Hospital in Savona (Italy) included an antero-posterior X-ray, a scaphoid-view X-ray, and a CT scan offering both 2D and 3D multiplanar reconstructions. These 3D reconstructions were instrumental in determining the quantity and location of bone shards.

### 2.4. Surgical Technique

A dorsal-radial approach was used for the partial scaphoid prosthesis, with a curved incision of approximately 5 cm medial to the course of the Extensor pollicis longus (EPL), which was isolated and retracted, as were the vascular structures and sensory branches of the radial nerve [15]. The extensor carpi radialis longus (ECRL) were mobilized and moved laterally to better expose the dorsal radiocarpal ligament, which was incised in an inverted T-shape and carefully dissected from the carpus. The entire scaphoid bone was exposed and isolated. After removing the necrotic proximal pole and the remaining tissue of the scapholunate ligament (SLL), the distal pole was preserved. A palmaris longus (PL) tendon graft was used to reconstruct the SLL. In one case, harvesting the PL tendon graft was impossible, so it was decided to stabilize the implant with Garcia Elias tenodesis using flexor carpi radialis (FCR) muscle as the graft [16,17,18]. The implant was then prepared for placement. First, a strip of PL tendon graft was fixed across the corresponding notch of the implant, which was placed in its space. The remaining PL tendon graft was stabilized with one or two anchors (1.8 mm in diameter and 5.9 mm in length) in base of the size of strip of PL on the lunate (Figure 3). Radiographic views were performed to confirm the position of the implant (Figure 4). The stability of the prosthesis and laxity ore tightness of ligament reconstruction was assessed by passively performing flexion and extension, as well as lateral and rotational movements, before reconstruction of the dorsal radiocarpal ligament (DRCL). The risk of overstuffing was not present because we had a custom-made product after the CT scan for each patient. In three cases, the dorsal capsule was reinforced with an extensor retinaculum flap because there was not tissue to close the capsule. A surgical drain was left in place for 24 h to prevent the occurrence of hematoma. In the end, a thermoplastic splint was applied to the wrist in 10° extension and was maintained for four weeks.

The surgical approach performed in total scaphoid prostheses was the same as in partial scaphoid prostheses. Upon opening the capsule, the entire necrotic scaphoid was removed except for a small volar portion of the distal pole, approximately 3 × 3 mm, which was left in place to preserve the insertion of the radioscaphocapitate (RSC) ligament. Therefore, a hole was drilled in the trapezium under radiographic evaluation. This hole was designed to accommodate the distal tip of the prosthesis stem, a key point to stabilize the implant. To reconstruct the SLL, Arthrex™ labral tape (Naples, FL, USA) was inserted into the lunate with an anchor (1.8 mm in diameter and 5.9 mm in length). The labral tape is perfect for all reconstructions due to its size and stiffness. The implant was then prepared for insertion. First, the two cords of the labral tape were pulled through the corresponding holes of the implant, then the distal tip of the prosthesis stem was inserted into the hole of the trapezium [7]. The implant was inserted into its space, and the two strings of the labral tape were finally knotted in a notch in the implant (Figure 5). The implant was then stabilized on both sides: distally by its stem tip inserted into the trapezium and proximally by the labral tape attached to the lunate. After the radiographic position was confirmed, the stability of the implant was tested by moving the wrist in all directions. In the end, a surgical drain was placed and left in place for 24 h to prevent the occurrence of hematoma, and a thermoplastic splint was applied to the wrist in a 10° extension for the partial scaphoid prosthesis, maintained for four weeks.

For both patient groups, a standard three-month rehabilitation protocol was attended. The rehabilitation protocol involves 6 weeks immobilized with a wrist brace. followed by ROM release in assisted flexion–extension, laser, ionophoresis (20 sessions). By the end the patients used the brace for 4 weeks only at night. Clinical and radiographic follow-ups were performed at one, three, and six months, as well as one year after the surgical procedure, to monitor the range-of-motion recovery and to assess the appropriate position of the prosthetic stem in the trapezium, as well as to reveal any evidence of resorption or cysts in the carpal bones and any signs of early implant displacement (Figure 6 and Figure 7).

### 2.5. Data Extraction

The assessed parameters were the same for both patient groups: carpal height ratio (CHR) measured by Youm’s method [7], radioscaphoid angle, wrist extension and flexion, radial and ulnar deviation of the wrist, grip strength and pinch strength by Jamar’s test [7] on the operated and contralateral wrist, the Visual Analogue Scale (VAS) (which measures pain intensity at rest and under load), the Disabilities of Arm, Shoulder, and Hand (DASH) score, and the Patient-Rated Wrist Evaluation (PRWE). The outcome parameters were the media of measurement. In particular, carpal height in centimeters, radio scaphoid angle, extension, flexion, ulnar and radial deviation in grade, grip and pinch strength in kilograms, and pain in the VAS scale.

To measure the carpal height on the X-rays, we used Radiant software (Version 8.0). To grade for radio scaphoid angle, extension, flexion, ulnar and radial deviation we used a protractor. To assess grip and pinch strength we used the Jamar instrument. To determine pain level, we used a VAS scale from 0 to 10 as indicated by patients.

### 2.6. Ethical Approval

The Institutional Review Board (IRB) of the University of Palermo defined this study as exempt from IRB approval (as a retrospective study of an established surgical procedure). All patients gave informed consent prior being included in the study. This study was conducted following the ethical standards outlined in the Declaration of Helsinki (1964) and its subsequent amendments.

### 2.7. Statistical Analysis

Descriptive statistical analysis was performed for all data extracted using R software, version 4.0.5 (2020; R Core Team, Vienna, Austria). Mean values with a measure of variability as a range (minimum–maximum) were calculated for continuous variables. Absolute number and frequency distribution were calculated for categorical variables.

## 3. Results

Nineteen patients were included in study. The average follow-up of all 19 patients included in the present study was 18 (12–18) months. The average age of the nine patients (seven males and two females) treated with partial scaphoid prostheses was 27.5 (18–37) years. The mean age of the 10 patients (eight males and two females) treated with total scaphoid prosthesis was 33.3 (26–41) years. The most common mechanism of injury was fall on an outstretched hand.

Clinical, functional, and radiographic outcomes were evaluated for both groups of patients before surgery, and at one, three, six months, and one year after surgery, except for DASH score and PRWE, which were assessed before surgery and one year after surgery.

The results for partial scaphoid prostheses are shown in Table 1, while the results for patients treated with total scaphoid prostheses are shown in Table 2.

One case of stem dislocation was found in both groups of patients at 30 days and 45 days after surgery for partial and total scaphoid prostheses, respectively. Removing the implant and performing a three-angle arthrodesis were necessary in all two cases.

## 4. Discussion

The main finding is that a custom-made 3D-printed partial or total titanium prosthesis is a viable solution for patients with scaphoid non-union and necrosis or scaphoid bone destruction in which previous conservative or surgical treatment has failed. Addressing non-unions of the scaphoid involves a range of strategies to restore wrist motion, tailored to the individual patient’s needs [19,20]. It is essential to carefully select the right preoperative plan and the correct size of the implant to ensure the stability of both the volar and dorsal ligaments, as well as the distal end of the prosthesis. During the dorsal approach, the volar ligament is maintained, while the dorsal ligament is reconstructed once the prosthesis is in place [21]. The effect of the prosthetic stem is akin to performing a scaphoid-trapezium–trapezoid fusion, redistributing the load from the radioscaphoid joint to the scaphoid–trapezius–trapezoid complex. Finally, to augment stability, the scapholunate interosseous ligament (SLIL) is reconstructed by integrating labral tape within the lunate, which is then securely fastened to the implant [2,21]. The analyses showed clinical, functional, and radiographic improvement in all outcomes analyzed in partial and total scaphoid prostheses. The best result obtained was from the VAS pain scale, with an improvement from preoperative to one year after surgery, both at rest and under load. One year after surgery, the DASH score and PRWE results were good, with a high satisfaction rate of patients at the end of follow-up. SLIL reconstruction in both procedures provided excellent stability and prevented a mid-carpal bone collapse in short- to medium-term follow-up. In the case of failure, more aggressive procedures, such as scaphoidectomy, three-angle arthrodesis (a procedure performed in the two failure cases), or PRC, may be considered an option [13]. Any scaphoid prosthesis must replicate the original shape of the bone as accurately as possible to minimize non-physiologic kinematics and wear. That objective is achievable with 3D customizing, the current method of obtaining a patient-specific implant. Several studies [22,23,24] have shown substantial interindividual differences in carpal bone shape and scaphoid movement patterns.

The strengths of the present study are that the surgeries were performed by a senior hand surgeon, with the same reproducible surgical procedure. Furthermore, a standardized rehabilitation protocol was performed for all patients, with close clinical and radiographic follow-ups to monitor the treated patients better.

This study also has several limitations to be taken into consideration. First, since it is a retrospective study, it lacks patient randomization. Second, more follow-ups and a higher number of patients in the study may be required to detect additional causes of failure; one cause of prosthesis failure is due to an incorrect indication for carpal ligament instability due to an untreated high-energy trauma to the wrist in acute instability. Finally, one has to consider the long time for the prosthesis fabrication process (about three weeks after sending MRI and CT scans) and the high cost of a custom-made prosthesis (about EUR 3000). Further studies with more extensive follow-ups and larger samples could confirm these obtained results.

### Future Direction

Further research is needed to establish standardized protocols for the treatment of scaphoid non-unions with custom-made 3D-printed protheses to optimize its effectiveness. Another important consideration in the treatment of scaphoid non-unions is patient selection. It is crucial to identify patients who are most likely to benefit from custom-made 3D-printed protheses treatment. Factors such as trauma severity, age, and comorbidities may influence surgery and patient outcomes. Custom-made 3D-printed titanium partial or total scaphoid prostheses and scaphoid interosseous ligament reconstruction is a promising therapeutic option for scaphoid non-unions, but its efficacy is mixed and dependent on patient characteristics and on the type and severity of the fracture. With continued research, custom-made 3D-printed scaphoid protheses may be a valuable add-on for the treatment of scaphoid non-unions.

## 5. Conclusions

Partial or total 3D-printed titanium scaphoid prostheses and scaphoid interosseous ligament reconstruction could be a reliable surgical procedure for treating scaphoid non-unions and when it is impossible to save the scaphoid bone in appropriately selected patients, respecting the precise anatomy of the individual patient and allowing for satisfactory clinical, functional, and radiographic results. Both prosthesis implant designs have shown clinical benefits in treating scaphoid non-unions. Proper indications between partial and total scaphoid prostheses are key to improved outcomes.

## Figures and Tables

**Figure 1 healthcare-11-03123-f001:**
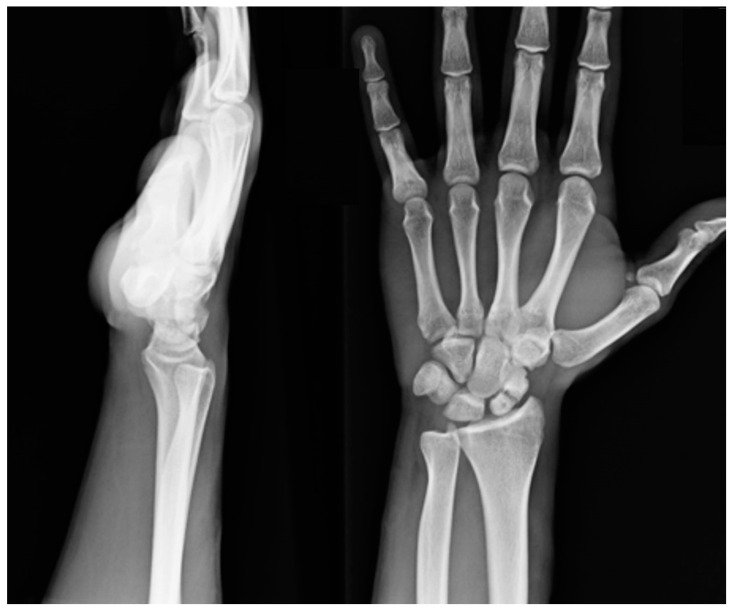
Preoperative radiographic imaging of a non-union case of the scaphoid proximal pole.

**Figure 2 healthcare-11-03123-f002:**
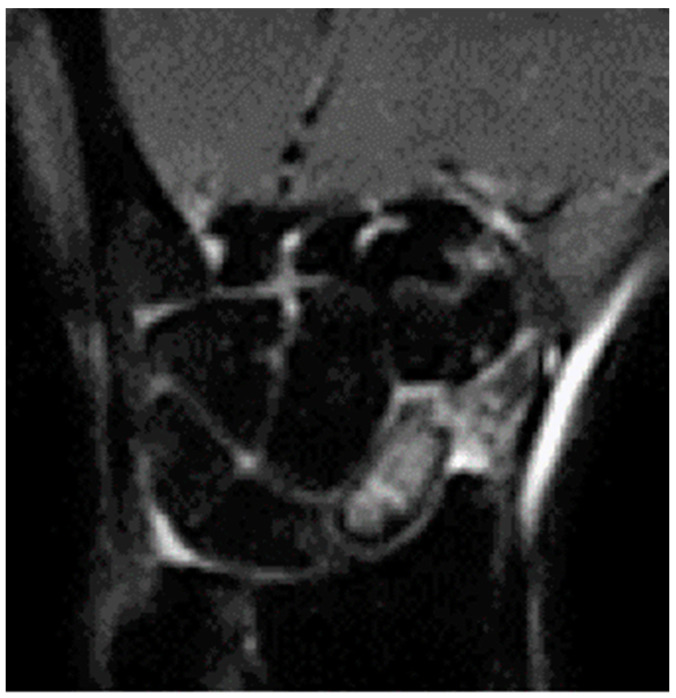
Preoperative MRI imaging of a non-union case of the proximal pole of the scaphoid with necrosis area.

**Figure 3 healthcare-11-03123-f003:**
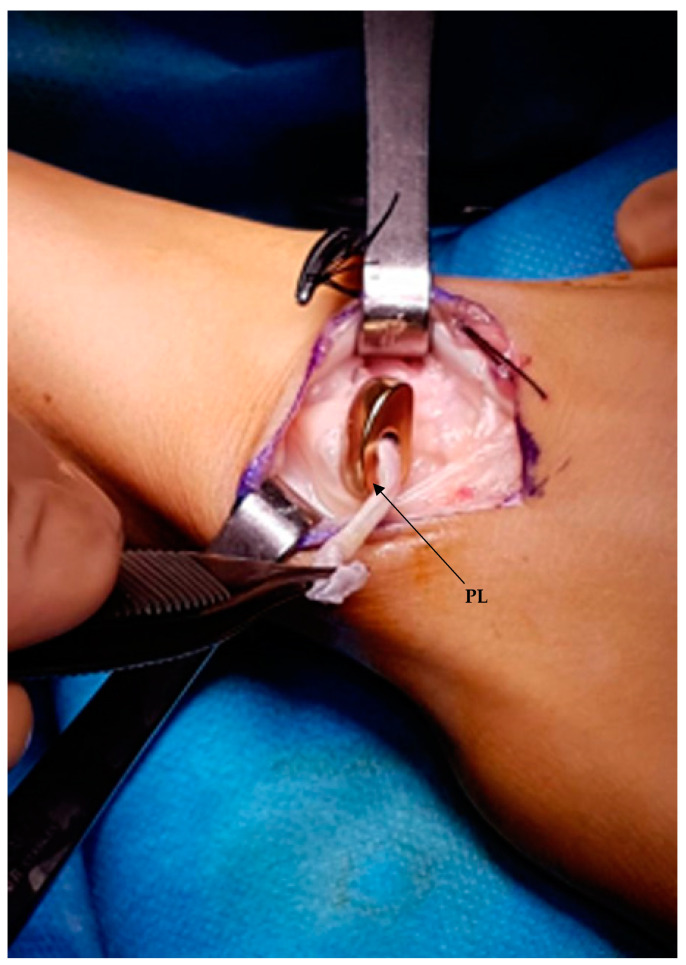
Intraoperative view. The custom-made 3D-printed titanium scaphoid prosthesis is placed; a palmaris longus (PL) tendon graft passing through the prosthetic body is attached to the lunate.

**Figure 4 healthcare-11-03123-f004:**
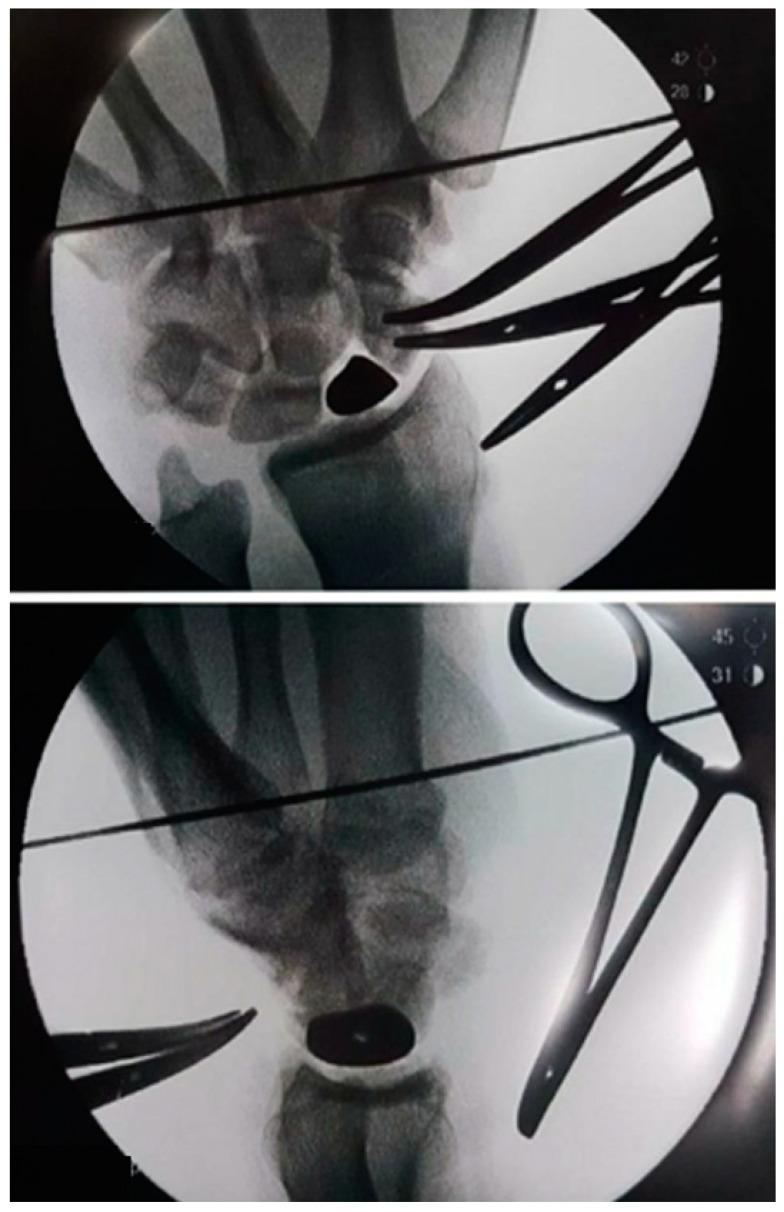
On perioperative radiography, the correct position of the scaphoid prosthesis was verified.

**Figure 5 healthcare-11-03123-f005:**
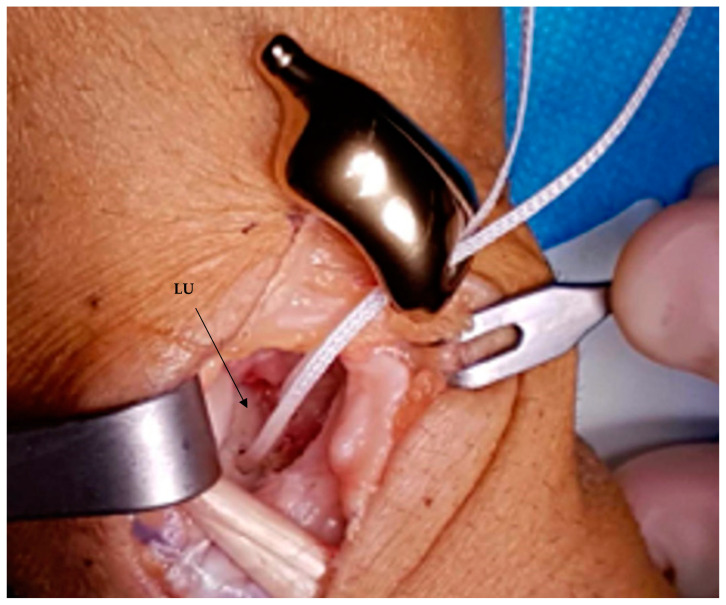
Intraoperative view. The custom-made 3D-printed titanium scaphoid prosthesis was placed. Arthrex™ labral tape was fixed in the lunate (LU) and passed through the implant. The distal stem of the implant was inserted into the prepared hole in the trapezium.

**Figure 6 healthcare-11-03123-f006:**
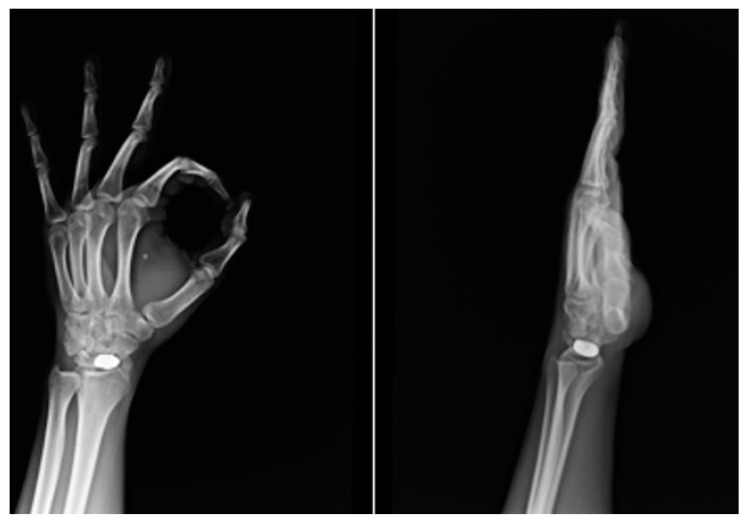
At one-year follow-up, radiography showed a correct radioscaphoid angle, no displacement or signs of carpal arthritis, or other problems.

**Figure 7 healthcare-11-03123-f007:**
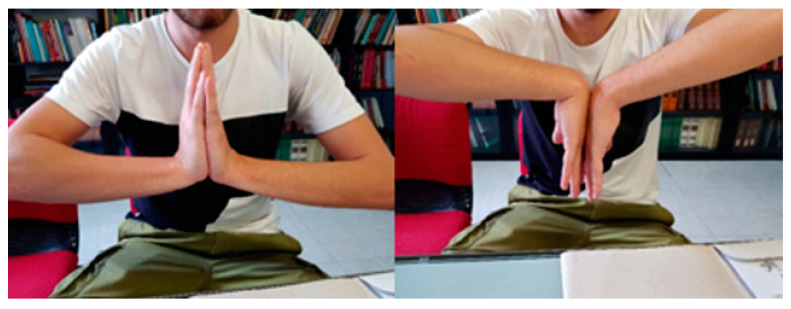
The wrist range of motion of a patient treated with a custom-made 3D-printed titanium scaphoid prostheses at a one-year follow-up.

**Table 1 healthcare-11-03123-t001:** Clinical, functional, and radiographic parameters of custom-made 3D-printed titanium partial scaphoid prostheses. °: degree; Kgf: kilogram-force.

Parameters	Pre-Operative, Mean (Min–Max)	1 Month after Surgery, Mean (Min–Max)	3 Months after Surgery, Mean (Min–Max)		6 Months after Surgery, Mean (Min–Max)	1 Year after Surgery, Mean (Min–Max)
Carpal height ratio (CHR)	0.84 (0.52–1.98)	0.77 (0.45–1.96)	0.77 (0.44–1.96)		0.76 (0.44–1.95)	0.76 (0.44–1.94)
Radioscaphoid angle, °	45.75° (37°–66°)	30.62° (20°–43°)	32.5° (30°–42°)		33.87° (30°–43°)	32.37° (30°–43°)
Wrist extension, °	33.57° (0°–70°)	53.57° (30°–65°)	57.14° (35°–70°)		63.57° (50°–80°)	64.28° (50°–80°)
Wrist flexion, °	40° (20°–70°)	41.42° (35°–50°)	47.85° (40°–60°)		59.28° (45°–70°)	60.71° (45°–70°)
Radial deviation of the wrist, °	12.85° (5°–35°)	17° (5°–22°)	21.42° (15°–30°)		26.42° (15°–60°)	26.45° (15°–60°)
Ulnar deviation of the wrist, °	27.85° (10°–45°)	36.14° (30°–50°)	39.28° (35°–45°)		37.14° (20°–50°)	36.42° (20°–50°)
Grip strength (Jamar’s test)						
Operated wrist, Kgf	17 (10–22)	12 (10–15)	15.6 (10–24)		24.7 (20–31)	29.1 (24–38)
Contralateral wrist, Kgf	39 (26–50)	37 (25–46)	37.1 (26–48)		36.4 (26–48)	30.8 (28–50)
Pinch strength (Jamar’s test)						
Operated wrist, Kgf	5.1 (4–6.5)	6.2 (3.5–8.5)	6.6 (3.5–8.5)		7.5 (6–8.9)	8 (7–9)
Contralateral wrist, Kgf	8.6 (10.5–10)	8 (6–9.5)	8.6 (6.5–9.5)		8.5 (6.5–9.5)	8.6 (6.5–9.5)
Visual Analogue Scale (VAS)						
At rest	3.6 (1–10)	1.9 (0–5)	0.4 (0–1)		0	0
Under load	7.4 (5–10)	4.4 (2–8)	3.1 (2–5)		2.5 (2–4)	1.7 (1–3)
Disabilities of Arm, Shoulder, and Hand (DASH) score	Pre–operative, Mean (Min–Max)			1 year after surgery, Mean (Min–Max)		
	22.4 (12.9–51)			9.2 (0.8–30.8)		
Patient Rated Wrist Evaluation (PRWE)	Pre–operative, Mean (Min–Max)			1 year after surgery, Mean (Min–Max)		
	33.7 (28–88)			17.5 (2–62)		

**Table 2 healthcare-11-03123-t002:** Clinical, functional, and radiographic parameters of custom-made 3D-printed titanium total scaphoid prostheses. °: degree; Kgf: kilogram-force.

Parameters	Pre-Operative, Mean (Min–Max)	1 Month after Surgery, Mean (Min–Max)	3 Months after Surgery, Mean (Min–Max)		6 Months after Surgery, Mean (Min–Max)	1 Year after Surgery, Mean (Min–Max)
Carpal height ratio (CHR)	0.61 (0.50–0.85)	0.59 (0.45–0.80)	0.58 (0.45–0.80)		0.53 (0.45–0.80)	0.53 (0.45–0.80)
Radioscaphoid angle, °	40.9° (12°–60.2°)	34.6° (44.5°–58°)	35° (32°–58°)		36.8° (30°–58°)	34.2° (30°–58°)
Wrist extension, °	31.4° (20°–55°)	42° (25°–65°)	47.8° (30°–70°)		47.1° (35°–70°)	52.1° (45°–75°)
Wrist flexion, °	25.7° (10°–45°)	36.7° (12°–50°)	44.2° (20°–60°)		44.2° (25°–60°)	47.8° (35°–60°)
Radial deviation of the wrist, °	10° (0°–25°)	13.5° (3°–25°)	16.8° (8°–25°)		20° (10°–30°)	28.5° (20°–45°)
Ulnar deviation of the wrist, °	29.7° (10°–60°)	29° (20°–35°)	29.7° (25°–35°)		32.1° (30°–45°)	35.5° (30°–45°)
Grip strength (Jamar’s test)						
Operated wrist, Kgf	13.6 (4–20)	15 (5–22)	18.8 (12–26)		22.4 (13–30)	25.7 (18–32)
Contralateral wrist, Kgf	29 (25–46)	28.7 (24–46)	34.7 (24–46)		37.5 (24–52)	36.5 (24–48)
Pinch strength (Jamar’s test)						
Operated wrist, Kgf	5 (1.8–10)	4.7 (2–12)	10.5 (2.5–16)		8 (5–16)	8.6 (10–16)
Contralateral wrist, Kgf	11.2 (7–22)	11.2 (6–22)	10.5 (6–22)		10.7 (6–22)	10.7 (6–22)
Visual Analogue Scale (VAS)						
At rest	5.5 (2–8)	1.8 (1–3)	1 (0–2)		0.8 (0–2)	0.1 (0–1)
Under load	8.2 (6–10)	4.7 (2–8)	3 (2–6)		3 (0–5)	1.4 (0–5)
Disabilities of Arm, Shoulder, and Hand (DASH) score	Pre–operative, Mean (Min–Max)			1 year after surgery, Mean (Min–Max)		
	28.2 (11.8–67.5)			11.8 (1.7–28.3)		
Patient Rated Wrist Evaluation (PRWE)	Pre–operative, Mean (Min–Max)			1 year after surgery, Mean (Min–Max)		
	35.6 (15–82)			22.5 (7–51)		

## Data Availability

The datasets used and/or analyzed during the current study are available from the corresponding author on reasonable request.

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
