# Peer review of "Treatment of Scaphoid Non-Unions with Custom-Made 3D-Printed Titanium Partial and Total Scaphoid Prostheses and Scaphoid Interosseous Ligament Reconstruction"

_healthcare, 2023, doi:10.3390/healthcare11243123_

Round 1

Reviewer 1 Report

Comments and Suggestions for Authors

The study aims to evaluate the clinical-functional and radiographic results after implanting custom-made 3D-printed titanium partial and total scaphoid prostheses and reconstruction of a scapholunate interosseous ligament. The study utilized a variety of clinical, functional, and radiographic outcome parameters to evaluate the performance of custom-made implants for both partial and total scaphoid prostheses and observed improvement in all outcome parameters. The novelty and significance of the study are high as it provides a reliable surgical procedure for treating scaphoid non-unions using patient-specific anatomy. The methodology of the study is sound, but the lack of in-depth statistical analysis weakens the conclusion and generalizability of the study.

The manuscript is well written with a clear and concise message. However, the manuscript needs to be further improved based on the following recommendations:

Author list:

·        Please use a different symbol to indicate the corresponding author and mark the corresponding author in the author list with that symbol

Introduction:

·        Line 48: Please remove “REF” and add a reference

·        Line 51: please put “3” in “[]” to mark it as a reference

·        Line 52: Please remove the word “be”

·        Line 55: Do the authors mean costly instead of cost-effective?

·        Lines 62-64: Please provide a reference for the presented information

·        Please provide a hypothesis for the study

Methods:

·        Line 115: No need to define “CT” again

·        Figures 3 & 5: Please label different components of the figure

·        Please cite and describe Fig 7 in the text

·        Please add a reference for Youm’s method and Jamar’s method.

·        Outcome parameters:

o   Please provide more information on how the outcome parameters were calculated/measured for reproducibility (e.g. equipment used, procedure, units of measurement, figures, equations, etc.)

o   In my opinion, it would be beneficial to the readers if the authors added a figure describing/showing how the functional and radiographic parameters were calculated

o   Please provide more information about the questionnaires. What did they evaluate? How many sections? Questions per section? Score range for each question? How a representative score was calculated for each subject (e.g. sum, average, etc.)? What do higher or lower values in the score range signify? etc.   

·        Please add a statement about informed consent from participants in Section 2.6

·        Please perform a statistical comparison across different time points for each outcome parameter (e.g. t-test, ANOVA, etc. or equivalent) to evaluate the improvement in patient outcomes. This will also allow the authors and the readers to understand the on-set of timepoint for improvement in patient outcomes

Results:

·        Tables 1 & 2: Poor resolution. Please replace it with a high-resolution figure. In my opinion, it would be better if the authors created a table instead of adding a figure of the table

Discussion:

·        Please compare and discuss the results of the proposed method with other studies using the same methodology, alternative techniques mentioned in the introduction, and healthy population data from the literature to understand the performance level of the proposed method. This will allow the authors to discuss the repeatability, reliability, and clinical significance of the proposed method, respectively.

·        Lines 230-243: Strong claims not backed up by data. Cannot claim significant improvement without performing statistical comparisons. The observed improvement might not be statistically significant

·        Line 247: Please replace “A” with “a”

·        Line 254: Do the authors mean a high number of patients instead of a low one?

References:

·        Please remove the duplicated numbers at the beginning of the references

Author Response

Introduction:

  • Line 48: Please remove “REF” and add a reference

Thank you done

  • Line 51: please put “3” in “[]” to mark it as a reference

Thank you done

  • Line 52: Please remove the word “be”

Thank you done

  • Line 55: Do the authors mean costly instead of cost-effective?

Thank you yes it has been modified

  • Lines 62-64: Please provide a reference for the presented information

Thank you a reference has been added

  • Please provide a hypothesis for the study

Thank you an hypothesis has been added (line 85-87)

Methods:

  • Line 115: No need to define “CT” again

Thank you done

  • Figures 3 & 5: Please label different components of the figure

Thank you done

  • Please cite and describe Fig 7 in the text

Thank you this has been added

  • Please add a reference for Youm’s method and Jamar’s method.

Thank you this has been added

  • Outcome parameters:

o   Please provide more information on how the outcome parameters were calculated/measured for reproducibility (e.g. equipment used, procedure, units of measurement, figures, equations, etc.)

Thank you yes it has been modified

o   In my opinion, it would be beneficial to the readers if the authors added a figure describing/showing how the functional and radiographic parameters were calculated

Thank you yes it has been modified

o   Please provide more information about the questionnaires. What did they evaluate? How many sections? Questions per section? Score range for each question? How a representative score was calculated for each subject (e.g. sum, average, etc.)? What do higher or lower values in the score range signify? etc. 

Thank you yes it has been modified

  • Please add a statement about informed consent from participants in Section 2.6

Thank you yes it has been modified

  • Please perform a statistical comparison across different time points for each outcome parameter (e.g. t-test, ANOVA, etc. or equivalent) to evaluate the improvement in patient outcomes. This will also allow the authors and the readers to understand the on-set of timepoint for improvement in patient outcomes

Thanks The timing of the improvement in patient outcomes can be seen in tables 1 and 2

Results:

  • Tables 1 & 2: Poor resolution. Please replace it with a high-resolution figure. In my opinion, it would be better if the authors created a table instead of adding a figure of the table

Thank you yes it has been modified

Discussion:

  • Lines 230-243: Strong claims not backed up by data. Cannot claim significant improvement without performing statistical comparisons. The observed improvement might not be statistically significant

Thank you yes it has been modified

  • Line 247: Please replace “A” with “a”

Thank you yes it has been modified

  • Line 254: Do the authors mean a high number of patients instead of a low one?

Thank you yes it has been modified

References:

  • Please remove the duplicated numbers at the beginning of the references

Thank you yes it has been modified

Reviewer 2 Report

Comments and Suggestions for Authors

The paper describes the Treatment of scaphoid non-unions with custom-made 3D- 2 printed titanium written in 13 pages contains 13 graphs, 2 tables and 17 references older than 5 years from 24 references (71%). The main aim of the study was satisfied. The methods of the study were chosen correctly but retrospectivly.  The abstract is constructed. Study do not brings new extra knowledge but conferm this method. Work meets the targets set.

The purpose and significance of the thesis of the article is stated clearly. The research study/review methods are sound and appropriate.

The topic is straightforward. The suitability of the chosen methods of processing solved problems is correct. The formulation of work objectives and the extent of their fulfilment are correct. The scope and level of achieved results were well achieved. The study provides a good analysis and interpretation of results and formulation of conclusions. Some other positives of this study are the usability of results in practice, transparency, and logical structure. The formal, linguistic, and stylistic level of work is very good.

The article title and abstract are appropriate.

Ethical approval wasn’t obtained, due to data being collected from records of the The hospital.

The authors have no conflicts of interest, including commercial interest, to disclose?

Comments on the Quality of English Language

The manuscript was written in UK English

Author Response

The authors have no conflicts of interest, including commercial interest, to disclose?

Thank you, the authors have no conflicts of interest, including commercial interest.

Reviewer 3 Report

Comments and Suggestions for Authors

Dear Authors,

a really interesting retrospective study in a hot topic. Concept and design was good but, as far as I am considered, two major flaws were detected: no statistical analysis of results was done while preoperative planning, patient selection and surgical procedures needs further improvement providing more details. Moreover, rehabilitation protocol is missing. Any treatment method should be reproducible and any complications should be analysed in order to become a viable option.

Line 3: Instead of "and" preferably use "with".

Line 40: Outstanding means great, fantastic. Was this the rationale?

Line 99: Please provide the time frame after the index injury and how the diagnosis of nonunion was confirmed.

Line 100: Please provide in details (for each participant) all previous surgical treatments which failed.

Lines 101-102: What do you mean with irreparable bone damage? How was confirmed that a surgical treatment was failed?

Line 117: As mentioned before, please provide in detail surgical approach, how prosthesis was attached to distal pole, seize of anchors, when one only and when two anchors, the rationale of dorsal capsule reinforcement by extensor retinaculum flap (lines 134-135), how risk of overstuffing was overcome, how ligament reconstruction regarding any laxity or tightness was intraoperatively evaluated as distal pole was not firmly attached (partial prosthesis). Please provide the same data for total scaphoid prosthesis, especially the use of labral tape (term "lunate" is preferable to "semilunar" line 150).

Line 165: Please provide in details the rehabilitation protocol as this is as important as the surgical procedure.

Line 178-183: Please provide definitions of these parameters, normal values, methods of evaluation, references and methods of these measurements/tests were performed. The use of an X-ray with all radiological values on could be very useful. 

Line 196: An average  follow-up period of 18 months with a range of 12-18 seems strange to me. Is this correct?

Lines 199-200: Please provide data for all mechanisms of injury apart from the most common. 

Lines 201-203: Please provide all the necessary statistical data proving that the two groups were comparable with similar homogeneity. Regarding table 1 and 2, statistical analysis with p-values with or without graphical presentation is mandatory. Postoperatively, contralateral grip strength was continuously reduced (table 1). Which could be the cause?

Lines 213-215: Which could be the cause of the complication? Please provide data regarding patients age, previous surgery, cause of injury.

Lines 247-249: Please clarify the meaning of the phrase.

Line 253: Please be specific.

Lines 254-255: Please mention some causes of failure as nothing was mentioned yet.

Lines 255-258: Cost-effectiveness analysis is lacking. Participants of this study are relative young and socio-economic active. What about return to work (same or different) ratio? Were they handworkers? Is there any contraindication of whole or partial scaphoid replacement apart from carpal instability and arthritis?

Author Response

Line 3: Instead of "and" preferably use "with".

Thank you yes it has been modified

Line 40: Outstanding means great, fantastic. Was this the rationale?

Thank you. The meaning of the sentence is that the sequelae of scaphoid fractures are still unclear.

Line 99: Please provide the time frame after the index injury and how the diagnosis of nonunion was confirmed.

Thank you, the diagnosis was done after 6 months and was confirmed by MRI

Line 100: Please provide in details (for each participant) all previous surgical treatments which failed.

Thank you. Patients with partial Prostheses: 9 out of 9 no treatment ab initio. Of the 10 with total prostheses: 5 no treatment (misdiagnosed), 3 with plaster cast for 2 months, 1 treated with k-wires removed after 6 weeks and 1 treated with ridge graft according to the Matti-Russe technique.

Lines 101-102: What do you mean with irreparable bone damage? How was confirmed that a surgical treatment was failed?

Thank you, The scaphoid can no longer be synthesised because it is necrotic, visualised by MRI

Line 117: As mentioned before, please provide in detail surgical approach, how prosthesis was attached to distal pole, seize of anchors, when one only and when two anchors, the rationale of dorsal capsule reinforcement by extensor retinaculum flap (lines 134-135), how risk of overstuffing was overcome, how ligament reconstruction regarding any laxity or tightness was intraoperatively evaluated as distal pole was not firmly attached (partial prosthesis). Please provide the same data for total scaphoid prosthesis, especially the use of labral tape (term "lunate" is preferable to "semilunar" line 150).

Thank you, The partial prosthesis is anchored to the distal scaphoid with a neo ligament and with 1 anchor of 1.6 mm. The total prosthesis has a distal stem that fits into a housing drilled with a 3 mm burr in the trapezium and the proximal semilunar with a neo ligament or with a gracile palmar homograft. The retinaculum flap of the extensors is made to cover the dorsal capsule after its suture.

Line 165: Please provide in details the rehabilitation protocol as this is as important as the surgical procedure.

Thank you, the rehabilitation protocol involves 6 weeks immobilised with a wrist brace. Followed by Roma release in assisted flexion-extension, laser, iontophoresis.

Line 178-183: Please provide definitions of these parameters, normal values, methods of evaluation, references and methods of these measurements/tests were performed. The use of an X-ray with all radiological values on could be very useful. 

Thanks, Everything was calculated on the basis of X-rays

Line 196: An average  follow-up period of 18 months with a range of 12-18 seems strange to me. Is this correct?

Thank you yes is correct

Lines 199-200: Please provide data for all mechanisms of injury apart from the most common.

Thnaks The most common mechanism of injury was fall on an outstretched hand.

 Lines 201-203: Please provide all the necessary statistical data proving that the two groups were comparable with similar homogeneity. Regarding table 1 and 2, statistical analysis with p-values with or without graphical presentation is mandatory. Postoperatively, contralateral grip strength was continuously reduced (table 1). Which could be the cause?

Thanks, This is due to intensive physiotherapy activities

Lines 213-215: Which could be the cause of the complication? Please provide data regarding patients age, previous surgery, cause of injury.

Thanks, The causes of the complications are due to an incorrect indication for carpal ligament instability due to a high-energy trauma to the wrist that was not treated acutely and existed in instability age 24

Lines 247-249: Please clarify the meaning of the phrase.

Thank you yes it has been modified

Line 253: Please be specific.

Thank you yes it has been modified

Lines 254-255: Please mention some causes of failure as nothing was mentioned yet.

      Thank you this has been added, one cause of prosthesis failure is due to an incorrect indication for carpal ligament instability due to an untreated high-energy trauma to the wrist in acute instability.

Lines 255-258: Cost-effectiveness analysis is lacking. Participants of this study are relative young and socio-economic active. What about return to work (same or different) ratio? Were they handworkers? Is there any contraindication of whole or partial scaphoid replacement apart from carpal instability and arthritis?

 Thank you, the return to work was the same. not all were manual workers. There are no contraindications apart from instability and arthritis

Round 2

Reviewer 1 Report

Comments and Suggestions for Authors

Thank you for diligently addressing all the comments and making appropriate changes to the manuscript. I just have a few minor recommendations:

1.     Figures 3 & 5: The labels are not legible. Please use an arrow and a different text color.

2.     Discussion: Please compare and discuss the results of the proposed method with other studies using the same methodology, alternative techniques mentioned in the introduction, and healthy population data from the literature to understand the performance level of the proposed method. This will allow the authors to discuss the repeatability, reliability, and clinical significance of the proposed method, respectively.

Author Response

  1. Figures 3 & 5: The labels are not legible. Please use an arrow and a different text color.

           Thank you yes it has been modified

  1. Discussion: Please compare and discuss the results of the proposed method with other studies using the same methodology, alternative techniques mentioned in the introduction, and healthy population data from the literature to understand the performance level of the proposed method. This will allow the authors to discuss the repeatability, reliability, and clinical significance of the proposed method, respectively.

Thank you.

To date, only a case report https://casereports.bmj.com/content/14/7/e241090 with the same methodology is reported in the literature but not studies with multiple patients.

In addition, I found a study analyzing Polyetheretherketone (PEEK) Biomaterial for scaphoid prosthesis but not useful for our comparison purpose https://www.hindawi.com/journals/bmri/2021/1301028/

Reviewer 3 Report

Comments and Suggestions for Authors

Dear Authors,

I have made some comments to improve the quality and the reliability of the original manuscript. Everything should be crystal clear to any reader regarding the surgical approach, the rehabilitation protocol and the clinical and radiological outcomes. Your treatment algorithm should be characterized by repeatability when applied by others. This is the scientific way. To do so, there is need to eliminate any dark or grey points. This is the role of the reviewer. You have answered to all my recommendations, but little was added on the manuscript. So I will repeat my comments point-by-point and I am waiting to see their answers written or mentioned on the manuscript in details when this is needed (surgical procedure, rehabilitation protocol, clinical and radiological outcomes etc.)

Line 3: Modified.

Line 15: This affiliation belongs to whom?

Line 41: Please change the term "outstanding". It is not scientifically correct. It has a subjective  meaning while the scientific language is objective.

Line 103: Please provide in details and written on the manuscript what was the time frame after the index injury and how the diagnosis of nonunion was confirmed.

Lines 104-105 : What do you mean with irreparable bone damage? Provide reference. How was confirmed that a surgical treatment was failed? Provide reference.

Line 106: No treatment means a misdiagnosed fracture or conservative treatment with cast. This is not clear for the 9 participants with partial prostheses.

Line 123 : Please modify the manuscript and " provide in detail surgical approach, how prosthesis was attached to distal pole, seize of anchors, when one only and when two anchors, the rationale of dorsal capsule reinforcement by extensor retinaculum flap (lines 138-139), how risk of overstuffing was overcome, how ligament reconstruction regarding any laxity or tightness was intraoperatively evaluated as distal pole was not firmly attached (partial prosthesis). Please provide the same data for total scaphoid prosthesis, especially the use of labral tape (term "lunate" is preferable to "semilunar" line 156).

Lines 172-173: Please provide IN DETAILS the rehabilitations protocol. When was doing what and why. Number of sessions etc.

Line 185: Please answer scientifically to my comment "Please provide definitions of these parameters, normal values, methods of evaluation, references and methods of these measurements/tests were performed. The use of an X-ray with all radiological values on could be very useful." This "Thanks, Everything was calculated on the basis of X-rays" is not scientifically acceptable.

Lines 211-212: Please form a table for your 19 participants where all preoperative and postoperative scores calculated are written, with the follow-up period. This table should also have data regarding previous surgeries, mechanism of injury and a patient-specific rehabilitation protocol. You can add this table on the manuscript or as a supplement file. 

Line 223: Please answer with a scientific way the following comment "Please provide all the necessary statistical data proving that the two groups were comparable with similar homogeneity. Regarding table 1 and 2, statistical analysis with p-values with or without graphical presentation is mandatory. Postoperatively, contralateral grip strength was continuously reduced (table 1). Which could be the cause?"

Lines 267-270: Please check the phrase. Unclear meaning. Also add some causes of failure and form a paragraph where cost-effectiveness analysis is added. Mention (in written) any contraindications with references.

Author Response

Line 3: Modified.

Line 15: This affiliation belongs to whom?

Thanks I have modified sorry

Line 41: Please change the term "outstanding". It is not scientifically correct. It has a subjective  meaning while the scientific language is objective.

Thank you I have changed with

Difficult

Line 103: Please provide in details and written on the manuscript what was the time frame after the index injury and how the diagnosis of nonunion was confirmed.

Thanks I have modified

 This diagnosis was done after CT scan and MRI, after 3 years in media from the injury (Line 103-104)

Lines 104-105 : What do you mean with irreparable bone damage? Provide reference. How was confirmed that a surgical treatment was failed? Provide reference.

Thanks I have modified 

Patients with partial Prostheses: 9 out of 9 no treatment ab initio. Of the 10 with total prostheses: 5 no treatment (misdiagnosed), 3 with plaster cast for 2 months, 1 treated with k-wires removed after 6 weeks and 1 treated with ridge graft according to the Matti-Russe technique.(Line 106-110)

Line 106: No treatment means a misdiagnosed fracture or conservative treatment with cast. This is not clear for the 9 participants with partial prostheses.

Thanks

See line 106-110

Line 123 : Please modify the manuscript and " provide in detail surgical approach, how prosthesis was attached to distal pole, seize of anchors, when one only and when two anchors, the rationale of dorsal capsule reinforcement by extensor retinaculum flap (lines 138-139), how risk of overstuffing was overcome, how ligament reconstruction regarding any laxity or tightness was intraoperatively evaluated as distal pole was not firmly attached (partial prosthesis). Please provide the same data for total scaphoid prosthesis, especially the use of labral tape (term "lunate" is preferable to "semilunar" line 156).

Thanks I have modified

Line 123: It is clearly specified that the distal pole of the scaphoid partial prostheses is preserved (line 133).

(1.8 mm in diameter and 5.9 mm in length) in base of the size of strip of PL (line 139-140)

The stability of the prosthesis and laxity ore tightness of ligament recostruction was assessed by passively performing flexion and extension, lateral and rotational movements, before reconstruction of the dorsal radiocarpal ligament (DRCL). The risk of overstuffing was not present because we had a custome made product after CT scan for each patient (line from 141 to 145)

In three cases, the dorsal capsule was reinforced with an extensor retinaculum flap because there was not tissue to close the capsule (line 145-146)

To reconstruct the SLL, Arthrex™ labral tape (Naples, FL, USA) was inserted into the lunate with an anchor (1.8 mm in diameter and 5.9 mm in length).The labral tape it’s perfect for the all recostruction for the size and stifness (line from 162 to 164)

Lines 172-173: Please provide IN DETAILS the rehabilitations protocol. When was doing what and why. Number of sessions etc.

Thanks.

For both patient groups, a standard three-month rehabilitation protocol was attended.  The rehabilitation protocol involves 6 weeks immobilised with a wrist brace. followed by ROM release in assisted flexion-extension, laser, ionophoresis (20 sessions). At the end the patients had brace for 4 weeks only the night (Line from 179 to 182)

Line 185: Please answer scientifically to my comment "Please provide definitions of these parameters, normal values, methods of evaluation, references and methods of these measurements/tests were performed. The use of an X-ray with all radiological values on could be very useful." This "Thanks, Everything was calculated on the basis of X-rays" is not scientifically acceptable.

Thanks

Everything the editor is requesting is described from line 195 to line 207.

Lines 211-212: Please form a table for your 19 participants where all preoperative and postoperative scores calculated are written, with the follow-up period. This table should also have data regarding previous surgeries, mechanism of injury and a patient-specific rehabilitation protocol. You can add this table on the manuscript or as a supplement file. 

Thanks

The table is already provided (Table 1 and 2) with everything that is requested. The mechanism of injury is the same for all (road trauma), and the physiotherapy protocol is the same for everyone (see lines 179 to 182).

Line 223: Please answer with a scientific way the following comment "Please provide all the necessary statistical data proving that the two groups were comparable with similar homogeneity. Regarding table 1 and 2, statistical analysis with p-values with or without graphical presentation is mandatory. Postoperatively, contralateral grip strength was continuously reduced (table 1). Which could be the cause?"

Thanks

Already described (see lines 214 to 218). The cause of the decreasing force in the contralateral limb is likely attributed to overuse due to physiotherapy.

Lines 267-270: Please check the phrase. Unclear meaning. Also add some causes of failure and form a paragraph where cost-effectiveness analysis is added. Mention (in written) any contraindications with references.

Thanks

Already described in the paragraph 4.1 and 5.